# PD-L1 Testing and Squamous Cell Carcinoma of the Head and Neck: A Multicenter Study on the Diagnostic Reproducibility of Different Protocols

**DOI:** 10.3390/cancers13020292

**Published:** 2021-01-14

**Authors:** Simona Crosta, Renzo Boldorini, Francesca Bono, Virginia Brambilla, Emanuele Dainese, Nicola Fusco, Andrea Gianatti, Vincenzo L’Imperio, Patrizia Morbini, Fabio Pagni

**Affiliations:** 1Section of Pathology, Department of Medicine and Surgery, University Milan Bicocca, ASST Monza, 20100 Milan, Italy; s.crosta2@campus.unimib.it (S.C.); polletta.79@libero.it (F.B.); virginia.brambilla.vb@gmail.com (V.B.); vincenzo.limperio@gmail.com (V.L.); 2Division of Pathology, Department of Health Sciences, University of Eastern Piedmont, 28100 Novara, Italy; renzo.boldorini@med.uniupo.it; 3Department of Pathology, ASST Lecco, 23900 Lecco, Italy; e.dainese@asst-lecco.it; 4IEO, European Institute of Oncology IRCCS, 20100 Milan, Italy; nicola.fusco@unimi.it; 5Pathology Unit, ASST Papa Giovanni XXIII, 24121 Bergamo, Italy; agianatti@asst-pg23.it; 6Pathology Unit, Policlinico S. Matteo, 27100 Pavia, Italy; patrizia.morbini@unipv.it

**Keywords:** PD-L1, head and neck carcinoma, pembrolizumab, HNSCC

## Abstract

**Simple Summary:**

The introduction of therapies with immune checkpoint inhibitors targeting the programmed cell death protein 1 and its ligand (PD-L1) axis in head and neck squamous cell carcinoma prompted the need of reliable bio-selectors to stratify patients that would benefit from these treatments. The assessment of PD-L1 expression through immunohistochemistry represents the most widely used method to perform this task, being recently approved by regulatory authorities. However, borrowing from previous experiences in lung cancer, the heterogeneity of antibodies and platforms used in the routine clinical practice requires a strict multi-institutional harmonization effort. In this setting, the present study is aimed to assess the performances of different PD-L1 staining protocols and the inter-observer reliability for its interpretation.

**Abstract:**

Immune checkpoint inhibitors for blocking the programmed cell death protein 1 (PD-1)/programmed death-ligand 1 (PD-L1) axis are now available for squamous cell carcinoma of the head and neck (HNSCC) in relapsing and/or metastatic settings. In this work, we compared the resulting combined positive score (CPS) of PD-L1 using alternative methods adopted in routine clinical practice and determined the level of diagnostic agreement and inter-observer reliability in this setting. The study applied 5 different protocols on 40 tissue microarrays from HNSCC. The error rate of the individual protocols ranged from a minimum of 7% to a maximum of 21%, the sensitivity from 79% to 96%, and the specificity from 50% to 100%. In the intermediate group (1 ≤ CPS < 20), the majority of errors consisted of an underestimation of PD-L1 expression. In strong expressors, 5 out of 14 samples (36%) were correctly evaluated by all the protocols, but no protocol was able to correctly identify all the “strong expressors”. The overall inter-observer agreement in PD-L1 CPS reached 87%. The inter-observer reliability was moderate, with an ICC of 0.774 (95% CI (0.651; 0.871)). In conclusion, our study showed moderate interobserver reliability among different protocols. In order to improve the performances, adequate specific training to evaluate PD-L1 by CPS in the HNSCC setting should be coordinated.

## 1. Introduction

Immune checkpoint inhibitors for blocking the programmed cell death protein 1 (PD-1)/programmed death-ligand 1 (PD-L1) axis are now available for squamous cell carcinoma of the head and neck (HNSCC) in relapsing and/or metastatic settings [1,2]. The introduction of this therapeutic strategy represents a valid alternative to the standard chemotherapy in recurrent/metastatic and non-resectable disease, significantly improving patients’ survival and reducing chemo-toxicity [3]. Such as in other tumor types, the identification of HNSCC patients eligible for the anti-PD-1 compound pembrolizumab is biomarker-based [4]. This is performed through the evaluation of PD-L1 expression by immunohistochemistry (IHC). However, the marked technologic and analytic heterogeneity related to the assessment of PD-L1 expression, coupled with the novelty of this bio-selector in HNSCC, represents a diagnostic challenge for pathologists [5,6,7]. 

A first issue is represented by the diverse commercially available antibody clones directed towards different epitopes of the intra- or extracellular portion of PD-L1. These different clones show a remarkable variability in terms of immunoreactivity (i.e., cytoplasmic and/or membrane), labeled cells (i.e., tumor and/or immune cells), patterns of expression (i.e., homogeneous or granular), and staining intensity [8,9]. To date, the most used anti-PD-L1 antibody clones are the 22C3 (Dako/gilent), which is certified as companion diagnostic test (CDx) for pembrolizumab in HNSCC; SP263; and SP142 (Ventana/Roche). These antibodies can potentially be employed with different staining platforms (i.e., Dako Autostainer Link 48, Dako Omnis, Ventana BenchMark Ultra, Leica Bond). The use of locally developed laboratory tests (LDTs) instead of the more expensive CDx requires harmonization for both pre-analytic (i.e., sample type, storage, space-time heterogeneity) and analytic factors (i.e., different evaluation algorithms and positivity cut-offs, subjective interpretative variability). This heterogeneity in terms of antibodies/platforms in the anatomic pathology departments has the advantage to confer versatility in the assessment of PD-L1 expression in different tumor types and organs (e.g., lung cancer) but requires an appropriate validation for their employment in different clinical settings. Moreover, after the pre-analytical phase, the degree of interobserver reproducibility of PD-L1 assessment could represent a further obstacle, particularly in HNSCC. This is due to the different scoring systems available for PD-L1, such as the tumor proportion score (TPS), the immune cell (IC) score, and the combined positive score (CPS) [4], the latter being recommended in the HNSCC setting [5].

In this scenario, establishing the degree of diagnostic concordance among the various methods is crucial to confer adequate reproducibility, with all the consequent therapeutic implications. In this work, we sought to determine the level of agreement for PD-L1 CPS in HNSCC samples by comparing the results obtained using different routine protocols.

## 2. Materials and Methods

### 2.1. Patients and Tissue Specimens

This multicenter observational retrospective study was approved by the Ethics Committee of the ASST Monza. The series included 15 histologic samples from surgical resections of HNSCCs collected from 1 January 2018 to 31 December 2019 at ASST Monza and was composed of specimens from the oral cavity (*n* = 8, 53%), pharynx (*n* = 1, 7%), and larynx (*n* = 6, 40%), eventually associated with metastatic dissected neck lymph nodes. Patients undergoing neoadjuvant chemotherapy and/or radiotherapy were not included in this study.

### 2.2. Tissue Microarrays (TMA) Construction

All the retrieved cases were represented by surgical specimens and for each case representative Hematoxylin and eosin (H&E)-stained slides were reviewed by two pathologists with experience in head and neck pathology (FP/FB) to identify the most representative areas for each tumor. Fixation of the surgical specimens was performed using 10% buffered formalin with an exposure that ranged from 12 to 48 h. Blocks were stored at room temperature and retrieved after the selection of the region of interest by the pathologists. These formalin-fixed, paraffin-embedded (FFPE) tissue blocks of the cases included in the study were then used to generate 2 TMAs. For each tumor, 2 representative tissue cores of 2 mm diameter were punched from the FFPE block and used to build the TMA. Areas enriched in tumor cells and peri-tumor inflammatory infiltrate were preferred, excluding necrosis or artifacts. TMAs were created with the ISE Galileo TMA R4.30 software and the ISE Galileo TMA CK4500 semi-automatic arrayer, produced by Integrated Systems Engineering S.r.l, (Milan, Italy).

### 2.3. Immunohistochemistry

We cut 3 μm thick sections from the TMA blocks and used them to perform IHC with anti-human pre-diluted antibodies against PD-L1. This analysis was performed using the gold standard (GS) protocol approved as CDx in the KEYNOTE 048 study (22C3, PharmDX on Autostainer Link 48) [1] and 5 alternative protocols, as detailed in Table 1. Slides obtained from TMAs were stained following each protocol in the different centers that participated in the study. The obtained immunostained slides were scanned from each center and then uploaded on the digital repository Spectrum (Leica Biosystems) to be assessed by a reference pathologist from each participant institution. Subsequently, the CPS was determined as the number of PD-L1-positive tumor cells, lymphocytes, and macrophages divided by the total number of viable tumor cells, multiplied by 100. Any perceptible and convincing partial or complete linear membrane staining of viable tumor cells that were perceived as distinct from cytoplasmic staining was considered as positive PD-L1 staining and was included in the scoring. Likewise, any membrane and/or cytoplasmic staining of mononuclear inflammatory cells within tumor nests and/or adjacent supporting stroma was considered positive PD-L1 staining and was included in the CPS numerator. Neutrophils, eosinophils, plasma cells, and ICs associated with in situ components, benign structures, or ulcers were excluded from the CPS score [10]. Although the result of the calculation can exceed the absolute value of 100, the maximum score was defined as CPS 100. The cases were then divided into PD-L1-negative (score 0, CPS < 1), intermediate expressors (score 1, 1 ≤ CPS < 20), and strong expressors (score 2, CPS ≥ 20).

### 2.4. Statistical Analyses

For the statistical analysis, each core of the TMAs was considered independent from all the others, including those coming from the same donor block and therefore from the same tumor. For pairs of evaluable cores from the same donor block, the percentage of concordance for each protocol in classifying them both as negative or positive (Concordance 1) or as negative, intermediate, or strong expressors (Concordance 2) was calculated. For each PD-L1 staining protocol, the sensitivity, specificity, positive predictive value, negative predictive value, and two types of error rates were calculated comparing each protocol to the GS. The first error (ER1) rate took into account only the number of false-positive (FP) and false-negative (FN) cases, while the second (ER2) included misclassifications as intermediate or strong expressors. The inter-observer agreement for the PD-L1 test was calculated through the overall percent agreement. The inter-observer reliability for the CPS score was determined by calculating the intraclass correlation coefficient (ICC) [11]. On the basis of a two-ways model, using the lower limit of the 95% confidence interval (CI), we defined ICC as poor (95% CI < 0.5), moderate (0.5 ≤ 95% CI < 0.75), good (0.75 ≤ 95% CI < 0.9), or excellent (95% CI ≥ 0.9). Moreover, the Fleiss’ kappa (κ) [12], which allows for the assessment of inter-observer reliability between two or more independent observers when the evaluation method is based on categorical variables, was tested (κ >0.8 strong agreement, 0.61 ≤ κ < 0.8 substantial, 0-41 ≤ κ < 0.6 moderate, 0.21 ≤ κ < 0.4 low). All statistical analyses were performed using Microsoft Excel 2013 and R software version 3.5.1 (R Foundation for Statistical Computing, Vienna, Austria).

## 3. Results

The study included 40 cores divided into 2 TMAs, taken from 15 patients (14 males, 1 female) with an average age of 67.5 years. The samples were derived from 15 primary tumors and 5 corresponding lymph node metastases. After the slides were stained, some cores were excluded for the absence of invasive tumoral components (e.g., fibro-muscular tissue or exclusive presence of carcinoma in situ). Excluded cases (*n* = 10) were classified as *unsatisfactory* by more than one protocol (2, 6, 7, 11, 19, 21, 26, 27, 31—Appendix A) or not assessable by the GS (39), obtaining 30 final cores on which the statistical analysis was performed (Table 2). Concordance assessed for each protocol and for the GS on paired cores from the same block/tumor are reported in Table 3. The high variability noted among the concordance percentages among different protocols and GS could be the expression of possible analytical (different TMAs) and biological (independent areas) heterogeneity of PD-L1 expression. The distribution of CPS scores with different protocols and with the GS is reported in Table 4. Considering the results with the GS, 80% of samples were positive, of which 47% were strong expressors (*n* = 14). This data appears in line with the results of the KEYNOTE-048 trial, which reported an expected prevalence of PD-L1 expression of 85% (43% strong expressors) [3]. False-negative (FN) and false-positive (FP) results were evaluated to determine the errors (ER1 and ER2) as well as the diagnostic performances for the different protocols (Table 5). Three out of six negative samples (50%) and 17 out of 24 positive samples (71%) were correctly evaluated by all the protocols. Including cases 13, 18, and 35, wrongly classified by a single protocol, the percentage of agreement was raised to 77%.

The ER1 of the individual protocol ranged from a minimum of 7% to a maximum of 21%, the sensitivity from 79% to 96%, and the specificity from 50% to 100%. The best performance was obtained using the SP142 antibody clone on the Ventana Benchmark Ultra platform (Protocol 2, ER1 = 7%; Sn = 92%; Sp = 100%). However, the presence of two not-evaluable cases (no. 37 and no. 38), classified as negative by the GS, could have slightly affected the diagnostic performances for this protocol. Table 6 reports the comparison of the CPS score obtained with each protocol with the results coming from the GS. In the negative group (CPS = 0), discrepancies were noted in three cases, one of them (no. 24) with discordant results for the majority of the protocols employed. In this case, the misclassification was due to the overestimation of staining in the context of punctate necrosis, which should not be evaluated as per interpretation criteria, as depicted in Figure 1. In the intermediate group (CPS = 1), 6 out of 19 errors were derived from the overestimation of the CPS score, whereas the majority were due to an underestimation. There were no cases in which different protocols over and underestimated alternatively. In the strong expressors group (CPS = 2), 5 out of 14 samples (36%) were correctly evaluated by all the protocols, but no protocol was able to correctly identify all the “strong expressors”. Some cases were underscored using the majority of the protocols, and one case (no. 34) was completely misinterpreted as negative due to a very focal and fairly perceptive staining (Figure 2). The inter-observer agreement for PD-L1 testing demonstrated an overall percent that reached the value of 87%.

The study demonstrated a moderate inter-observer reliability for the CPS score, with an ICC of 0.774 (95% CI (0.651; 0.871)). The inter-observer reliability assessed with the Fleiss’ kappa (κ) among different CPS score groups was moderate for the negative (κ = 0.43) and strong expressors (κ = 0.557) and fair for the intermediate group (κ = 0.303).

## 4. Discussion

The introduction of PD-1/PD-L1 inhibitors for the treatment of patients with relapsed and/or metastatic HNSCC raises the important issue of the reliability and reproducibility of the theranostic evaluation of PD-L1 expression. Few studies have evaluated the interobserver agreement among different pathologists for the PD-L1 assessment in HNSCC. The Food and Drug Administration (FDA) approved the 22c3 clone on the Autostainer platform as CDx on the basis of the demonstration of an overall percent agreement ranging from 95.7% to 97.8% in evaluating CPS > 1 and 92.1% to 97.3% for CPS > 20 [13]. However, the large employment of different tools for the purpose stimulated the comparison of diagnostic performances among different protocols, leading the College of American Pathologists to suggest a minimum sensitivity and specificity of 90% independently from the employed test [14]. Some reports showed a high overall percent agreement using SP263 (96.3%, with a positive cut-off equal to 25% tumoral cells) and SP142 clones (94% with a cut-off equal to 5% of ICs) [15,16]. A third study reported values between 73% and 98% from applying different positivity scores and cut-offs for each antibody clone used [9]. Recently, a large study demonstrated that the SP263 clone stained a higher percentage of cells, especially when the CPS was employed [17]. Moderate concordance was shown between three different PD-L1 assays, and considerable differences in PD-L1 positivity were observed when using clinically relevant cut-offs. The extreme heterogeneity of the anatomic pathology landscape in terms of available antibodies and platforms (and the consequent combination of these) could significantly impact on the stratification of patients candidate to immunotherapy. As a consequence, an in-depth evaluation of the current situation to verify the reliability of PD-L1 staining as a bio-selector for HNSCC in the clinical practice is mandatory [18,19,20,21].

In this TMA-based study, we compared the performances of the GS, as proposed by the KEYNOTE 048 study (22c3, PharmDx on Autostainer Link 48) and five different protocols. The results showed a specificity of 100% for the SP142 clone on the Ventana platform, while the SP263 clone showed the best performance in terms of sensitivity (96%). The remaining 22C3 antibody clone-based LDTs gave comparable good results in terms of specificity (80–83%) and variable sensitivity, ranging from 79 to 88%. The error rate in classifying the samples as positive or negative appeared low—from 7% up to 21%. In the SP142-based protocol, immunohistochemical results were formulated using reference ranges; this was due to the peculiar granular pattern of reactivity that characterizes the ICs, which discourage a precise CPS formulation.

As previously and largely demonstrated in lung pathology, the origin of the errors is intrinsically related both to interpretative variables (pathologists’ experience, appropriate training in CPS formulation) and to technical components (analytical issues, background noise, biological heterogeneity in the expression of PD-L1 among serial sections obtained from the same TMA core) [22,23,24]. The study demonstrated a good overall inter-observer agreement (87%), although this measure does not take into account the random agreement component that could potentially overestimate the actual agreement among observers. For this reason, the employment of an unbiased method (e.g., Fleiss’ kappa) could adequately assess this parameter, showing higher reproducibility for positive cases than negative ones. The interobserver reliability assessed by ICC for the CPS score was moderate (equal to 0.774), while the Fleiss’ kappa (κ) differed among CPS score groups, being moderate for the negative (κ = 0.43) and strong expressor (κ = 0.557) categories and low for the intermediate group (κ = 0.303). The lower reliability demonstrated with the intermediate expressors could have been due to the intrinsic difficulties encountered in the interpretation of these specimens, especially if represented by cases with a CPS score close to the range cut-offs (e.g., slightly higher than 1 or lower than 20). This could lead to the under/overestimation of the effective CPS value, as demonstrated in the present work (Table 6). In this setting, however, this distinction could have little if no consequences from the theranostic point of view but could preserve a prognostic role, as demonstrated by the results of the KEYNOTE 048 trial reporting a better response to immunotherapy treatment for patients with CPS ≥ 20 compared to those with CPS ≥ 1.

## 5. Conclusions

A few months after the European Medicines Agency (EMA) approval of the treatment with pembrolizumab for cases of relapsed and/or metastatic HNSCC, we in our feasibility study analyzed the different protocols and interobserver reliability in PD-L1 testing. On the basis of these preliminary findings, we found that intermediate expressors demonstrated the greatest interpretative difficulties, suggesting that adequate specific training to evaluate PD-L1 in the HNSCC setting by CPS could be potentially useful. Minor limitations of this work are the low number of cases analyzed and the relative under-representation of some specific head and neck districts (e.g., pharyngeal cancer). However, further studies on larger cohorts could benefit from this experience for the definitive standardization and validation of protocols different from the companion diagnostic, as well as ensuring diagnostic and interpretative harmonization of the results for the achievement of the best routine clinical management of these patients.

## Figures and Tables

**Figure 1 cancers-13-00292-f001:**
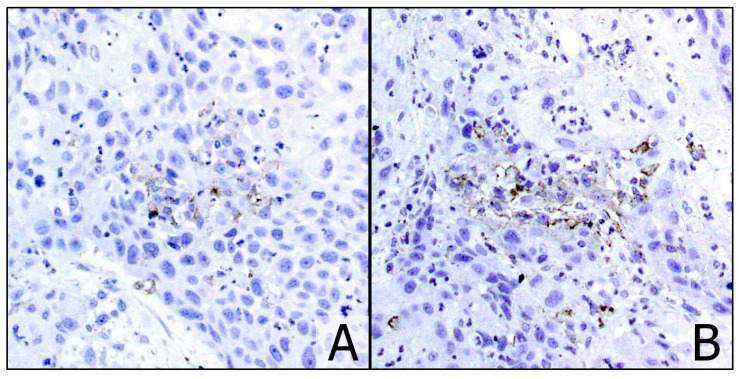
Case no. 24: overestimation of the staining in punctate necrosis areas (**A**) ((**B**) ×20). The case was classified as CPS 0 with the GS.

**Figure 2 cancers-13-00292-f002:**
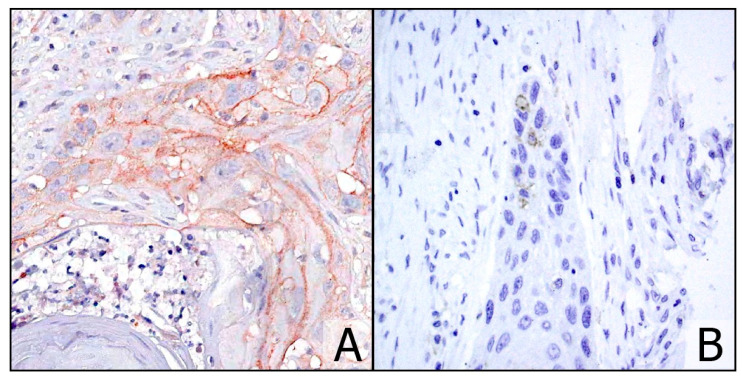
Case no. 34 with CPS = 2 using the GS (**A**) (×20) and interpreted as negative with protocol 1, although faint and very focal staining was present in restricted areas (**B**) (×20).

**Table 1 cancers-13-00292-t001:** Immunohistochemical protocols of the study.

Protocol	Clone	Platform
Protocol 1	22C3	Ventana Benchmark
Protocol 2	SP142	Ventana Benchmark
Protocol 3	22C3	Leica Bond
Protocol 4	SP263	Ventana Benchmark
Protocol 5	22C3	Dako, Omnis
Gold standard	22C3 pharmdx	Dako, Autostainer

**Table 2 cancers-13-00292-t002:** Combined positive score (CPS) values for the final statistical analysis. ne, not evaluable.

Case	1	3	4	5	8	9	10	12	13	14	15	16	17	18	20	22	23	24	25	28	29	30	32	33	34	35	36	37	38	40
Protocol 1	2	0	0	1	2	2	2	2	1	1	1	1	0	0	0	2	1	1	2	2	1	1	1	1	0	1	1	0	0	1
Protocol 2	2	1	0	2	2	2	1	2	1	1	2	2	0	0	0	2	1	0	2	2	1	1	2	2	1	0	1	ne	ne	1
Protocol 3	2	0	0	2	1	2	1	2	1	1	1	1	0	0	1	2	0	2	2	2	1	1	2	1	0	1	1	0	0	0
Protocol 4	2	1	0	2	2	1	1	2	1	1	1	1	0	1	1	2	0	2	2	2	2	1	2	2	1	1	1	0	1	1
Protocol 5	2	0	ne	2	2	2	1	2	0	1	1	2	0	0	0	2	0	2	2	2	1	1	2	2	1	1	2	0	0	0
GS	2	1	0	1	2	2	2	2	1	2	2	1	0	0	1	2	1	0	2	2	1	1	2	2	2	1	2	0	0	1

**Table 3 cancers-13-00292-t003:** Concordance assessed for each protocol and for the gold standard (GS) on paired cores from the same block/tumor.

	Protocol 1	Protocol 2	Protocol 3	Protocol 4	Protocol 5	GS
No. couples	14	13	14	14	13	17
Concordance 1	71%	77%	79%	86%	92%	82%
Concordance 2	43%	54%	64%	71%	85%	47%

Concordance 1 is related to the classification in positive or negative, concordance 2 refers to the classification in negative, intermediate, and strong expressors.

**Table 4 cancers-13-00292-t004:** Distribution of the CPS with different protocols.

CPS Score	0	1	2
Protocol 1	27%	47%	27%
Protocol 2	21%	36%	43%
Protocol 3	30%	40%	30%
Protocol 4	13%	50%	37%
Protocol 5	31%	24%	45%
GS	20%	33%	47%

**Table 5 cancers-13-00292-t005:** Diagnostic (technical and interpretative) performances for each protocol.

	Protocol 1	Protocol 2	Protocol 3	Protocol 4	Protocol 5
Sn	88%	92%	83%	96%	79%
Sp	83%	100%	83%	50%	80%
VPP	95%	100%	95%	88%	95%
VPN	63%	67%	56%	75%	44%
ER1	13%	7%	17%	13%	21%
ER2	30%	29%	40%	40%	41%

**Table 6 cancers-13-00292-t006:** Comparison of the CPS score for each protocol with the GS subdivided in negative, intermediate, and strong expressors. Red: false-positive and false-negative cases; Green: underestimation of intermediate expression; light blue: overestimated intermediate expressors. ne, not evaluable.

	CPS Negative as per GS						CPS Intermediate as per GS										CPS Strong as per GS													
Cases	4	17	18	24	37	38	3	5	13	16	20	23	29	30	35	40	1	8	9	10	12	14	15	22	25	28	32	33	34	36
Protocol 1	0	0	0	1	0	0	0	1	1	1	0	1	1	1	1	1	2	2	2	2	2	1	1	2	2	2	1	1	0	1
Protocol 2	0	0	0	0	ne	ne	1	2	1	2	0	1	1	1	0	1	2	2	2	1	2	1	2	2	2	2	2	2	1	1
Protocol 3	0	0	0	2	0	0	0	2	1	1	1	0	1	1	1	0	2	1	2	1	2	1	1	2	2	2	2	1	0	1
Protocol 4	0	0	1	2	0	1	1	2	1	1	1	0	2	1	1	1	2	2	1	1	2	1	1	2	2	2	2	2	1	1
Protocol 5	ne	0	0	2	0	0	0	2	0	2	0	0	1	1	1	0	2	2	2	1	2	1	1	2	2	2	2	2	1	2

## Data Availability

The data presented in this study are available in this article (and Appendix A).

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
