# Peer review of "PD-L1 Testing and Squamous Cell Carcinoma of the Head and Neck: A Multicenter Study on the Diagnostic Reproducibility of Different Protocols"

_cancers, 2021, doi:10.3390/cancers13020292_

Round 1

Reviewer 1 Report

It is a good idea to compare various protocols on PDL-1 IHC.

However this study needs major revisions:

  1. The language must be improved. I recommend to involve a native Speaker
  2. Only 15 histologic primary tumors were used and only 5 corresponding metastases were evaluated. The number needs to be increased.
  3. The sample collection is not representative since only one pharyngeal cancer is involved in contrast to 8 oral cavity and 6 laryngeal cancers.
  4. The authors evaluated the CPS score only. It would be interesting to evaluate the TPS score as well.

Author Response

Thanks a lot for the opportunity ro improve our paper, by your comments and suggestions.

REVIEWER 1 

It is a good idea to compare various protocols on PDL-1 IHC.

We are happy you appreciated our effort to compare different PD-L1 IHC protocols in the H&N district. We made our best to address all the suggestions of reviewers.

However this study needs major revisions:

  1. The language must be improved. I recommend to involve a native Speaker

ADDRESSED. We checked the text thanks to Prof Andrew Smith, BSC and professor of Biochemistry at UNIMIB. He come from UK, being English mother-tongue.

  1. Only 15 histologic primary tumors were used and only 5 corresponding metastases were evaluated. The number needs to be increased.

Thanks for your observation. However, we found that using a doubled series of 15+5 cases (for a total amount of 40 scorable cores) with 5 different staining protocols (+ the gold standard) would allow to obtain an adequate overview of the state of the art of the assessment of PD-L1 IHC in head and neck district. This is further supported by the presence of very recent reports (https://pubmed.ncbi.nlm.nih.gov/33275169/) that used an even lower number of cases (2-3) to assess the reproducibility of TPS in lung cancer. The main topic of our paper is a feasibility study on the application of the CPS for pathologists that are quite reluctant in introducing this tool in routine practice. We approved the request of the reviewer, reducing the conclusions of our job and showing the major limitations of the “state of the art” in this particular application of immunohistochemistry of our labs. Future larger trials may benefit for our opening results.

  1. The sample collection is not representative since only one pharyngeal cancer is involved in contrast to 8 oral cavity and 6 laryngeal cancers.

Thanks for your observation. We thought that having an equal representation of all the possible squamous cell carcinomas from each district in a same proportion would represent an added value for the study. However the similarity of squamous cell carcinomas arising in different anatomical sites for PD-L1 scoring was just demonstrated by the KEYNOTE 048 study. Since our test was mainly related to technical and interpretative findings and based on a TMA design, we thought that this was a minor limitation, we entered in the specific section of the manuscript.

Moreover squamous cell carcinoma (if not HPV/EBV-related) has potentially the same characteristics in every district and there are not (by now) known differences in PD-L1 assessment for SCC arising in different districts, the enrollment of an equal number of SCC per district is not strictly required. Finally, in terms of frequency oral and laryngeal cancers are by far the most frequent forms of head and neck cancers and so we tested the robustness of protocols and the reproducibility of PD-L1 assessment mainly in cancers arising in these two regions, that would be more useful for a practical point of view for the H&N pathologist.

  1. The authors evaluated the CPS score only. It would be interesting to evaluate the TPS score as well.

Thanks for your comment. However, the aim of the study is to assess the robustness of different protocols and the reliability of PD-L1 assessment in the head and neck cancers through the evaluation of combined positive score (CPS = N° PD-L1 positive cells/total cancer cells * 100), which has been recently introduced as a selector for the treatment with Pembrolizumab after the KEYNOTE 048 trial. Since the tumor proportion score (TPS = N° pos tumor cells/total tumor cells * 100) has not been validated for this aim, its assessment in the present paper would have been of less interest. Moreover, since the numerator of the CPS score does take into account ALSO the number of positive tumoral cells, our observations could be inferred to the TPS score as well. Moreover many different trials for NSCLC were recently published (including  the squamous cell carcinoma histotype) dedicated to the TPS performance evaluation. In our experience TPS is a robust and well-known tool, pathologists are familiar with. Since the main aim of the manuscript was the introduction of a new indicator like the CPS, we thought to focus the study on it.

Reviewer 2 Report

The study aims to analyze the inter-observer reliability of PD-L1 testing for HNSCC using 5 different protocols. 15 primary tumours (and 5 lymph node mets) were used and generated 40 TMAs, 10 cases were excluded. The samples underwent PD-L1 analysis using the golden standard from Keynote-048 as well as 5 other protocols. The results show a high specificity and sentisitivy overall but differences between the assays with some surprising differences in selected samples.

Major points:

Still somewhat unsure what was tested, the inter-observer reliability or the robustness of the protocols. The study is designed to test the protocols, but the conclusion is to increase PD-L1 evaluation-training?

Minor points:

The introduction is missing some key remarks on clinical impact, ie the cost-effect of having more than one protocol, the clinical impact on when treatment offers change for patients.

In the materials and methods the study is described as a "multicenter observational retrospective study", unclear what was done in different centers. Pathology evaluation or staining in different centers?

Also information on the tissue blocks is missing, fixation time, storage temp, certified laboratory, etc.

Table 2,  6 and S1 would benefit from additional information on abreviations (nv).

Line 171, intepretation of kappa should read "fair" not "low" for intermediate group. https://www.ncbi.nlm.nih.gov/books/NBK92287/table/executivesummary.t2/

The discussion would benifit if there was more on clinical implication, do the requirement for sensitivity and specitivity for assays fall below of what is required for FDA and CE-approval? 

Author Response

Thanks a lot for the opportunity to improve our study by your suggestions.

REVIEWER 2

The study aims to analyze the inter-observer reliability of PD-L1 testing for HNSCC using 5 different protocols. 15 primary tumours (and 5 lymph node mets) were used and generated 40 TMAs, 10 cases were excluded. The samples underwent PD-L1 analysis using the golden standard from Keynote-048 as well as 5 other protocols. The results show a high specificity and sentisitivy overall but differences between the assays with some surprising differences in selected samples.

Major points:

Still somewhat unsure what was tested, the inter-observer reliability or the robustness of the protocols. The study is designed to test the protocols, but the conclusion is to increase PD-L1 evaluation-training?

Thanks for your observation; our first version was confused. Actually, the aims of the study are both to assess the different performances of protocols available in the clinical practice as compared to the gold standard/companion diagnostic and, as a consequence, to assess the inter-observer agreement and reliability to score PD-L1 using the CPS required after the KEYNOTE experience. We re-edited the text so that both the aims of the study can be more clearly defined and discussed from introduction to the conclusions.

Minor points:

The introduction is missing some key remarks on clinical impact, ie the cost-effect of having more than one protocol, the clinical impact on when treatment offers change for patients.

On the base of your suggestion we added a sentence on the impact of anti PD-1 drugs in patients with HNSCC at the beginning of introduction: 

“The introduction of this therapeutic strategy represents a valid alternative to the standard chemotherapy in recurrent/metastatic and non-resectable disease, significantly improving patients survival and reducing chemo-toxicity”

and on the usefulness of having more than one antibody/platform/protocol in the AP lab:

“This heterogeneity in terms of antibodies/platforms in the anatomic pathology departments has the advantage to confer versatility in the assessment of PD-L1 expression in different tumor types and organs (e.g. lung cancer) but requires an appropriate validation for their employment in different clinical settings.”

In the materials and methods the study is described as a "multicenter observational retrospective study", unclear what was done in different centers. Pathology evaluation or staining in different centers?

We addressed this lacking information in the sub-section “Immunohistochemistry” specifying that:

 “Slides obtained from TMAs have been stained following each protocol in the different centers that participated in the study. The obtained immunostained slides have been scanned from each center and then uploaded on the digital repository Spectrum (Leica Biosystems) to be assessed by a reference pathologist from each participant institution.” 

Also information on the tissue blocks is missing, fixation time, storage temp, certified laboratory, etc.

ADDRESSED fixation time has been demonstrated not to affect the results of PD-L1 IHC https://pubmed.ncbi.nlm.nih.gov/31092453/

Table 2,  6 and S1 would benefit from additional information on abreviations (nv).

ADDRESSED

Line 171, intepretation of kappa should read "fair" not "low" for intermediate group. https://www.ncbi.nlm.nih.gov/books/NBK92287/table/executivesummary.t2/

ADDRESSED

The discussion would benifit if there was more on clinical implication, do the requirement for sensitivity and specitivity for assays fall below of what is required for FDA and CE-approval?

Based on your suggestion we modified the first part of discussion debating the differences in diagnostic performances among the results obtained by FDA during the approval procedure

Round 2

Reviewer 2 Report

No major issues to address, I’m satisfied with the response and changes.